# On the Need for Human Studies of PM Exposure Activation of the NLRP3 Inflammasome

**DOI:** 10.3390/toxics11030202

**Published:** 2023-02-21

**Authors:** Doug Brugge, Jianghong Li, Wig Zamore

**Affiliations:** 1Department of Public Health Sciences, School of Medicine, University of Connecticut, Farmington, CT 06030, USA; 2Institute for Community Research, Hartford, CT 06106, USA; 3Somerville Transportation Equity Partnership, Somerville, MA 02145, USA

**Keywords:** particulate matter air pollution, NLRP3 inflammasome, research needs

## Abstract

Particulate matter air pollution is associated with blood inflammatory biomarkers, however, the biological pathways from exposure to periferal inflammation are not well understood. We propose that the NLRP3 inflammasome is likely stimulated by ambient particulate matter, as it is by some other particles and call for more research into this pathway.

Research that we have helped lead, focused on associations of ultrafine particles (UFPs; particles with an aerodynamic diameter of < 100 nm) with inflammatory biomarkers in peripheral blood, has evolved naturally into an interest in research on inflammatory mechanisms [1,2,3]. In research on both UFP exposure and in the broader literature on particulate matter (PM) generally, especially on PM_2.5_ (PM of < 2.5 um), there are many reports of associations with biomarkers of inflammation, such as C-reactive protein (CRP) and interleukin-6 (IL-6) [4,5,6]. This has led to low-level, chronic systemic inflammation becoming a leading hypothesis for the underlying mechanism by which PM influence the development of adverse cardiovascular outcomes [7] as well as respiratory and neurological diseases [8].

The pathways that drive inflammatory responses are relatively well-elucidated (Figure 1). Because IL-6 promotes increases in CRP, we have long seen that associations of UFPs with both biomarkers in peripheral blood are quite similar. While our research teams have not studied the earlier steps in this pathway for UFPs, it is known that IL-6 levels are, in turn, stimulated by interleukin-1beta (IL-1beta), which has been activated by inflammasomes [9]. 

The NLRP3 inflammasome (nucleotide-binding oligomerization domain-like receptor (NLR) family pyrin domain-containing 3) is of special interest because it has been established that silica, uric acid, cholesterol crystals, and other particles stimulate NLRP3, leading to inflammation [11,12,13]. Since these forms of PM activate inflammation via NLRP3, it is possible that inflammation generated by ambient PM also acts through NLRP3. While there are many other inflammasomes, they respond highly specifically to infectious organisms. 

Thus, we hypothesize that UFP, PM_2.5_, and other ambient combustion PM are acting through NLRP3 to generate what is often called “sterile inflammation”; that is, inflammation without infectious agents, such as viruses or bacteria. The specific hypothesis would be that UFPs somehow activate NLRP3, which then generates IL-1beta and IL-18, leading to the production of IL-6 and, subsequently, CRP, both of which many have found to be associated with combustion PM. The resulting inflammation could then be associated with a wide range of diseases, including those that are cardiovascular, respiratory, and neurological.

In recent years, there has been increased laboratory-based research on PM and the activation of NLRP3, mostly in China. A cursory review of the literature turns up many relevant studies. A more systematic review would be warranted. 

Numerous studies in cell lines have reported the activation of NLRP3 and associated cytokines by PM_2.5_ [14,15,16,17,18,19,20] and, in one study, PM_10_ [21]. There has also been a similar output of studies on PM_2.5_ inducing NLRP3 in animal and rodent models [9,22,23,24,25,26,27,28]. A smaller number of reports suggested that there were factors that could reverse the activation of NLRP3 by PM [29,30]. Studies on UFPs are much rarer so far, with one study that we found reporting that black carbon nano PM could activate NLRP3 [31].

It is notable how recently these studies appeared, and it is likely that many more are in the pipeline. It is also critical to consider that, to our knowledge, the epidemiology of PM, and especially UFPs, effects on NLRP3 in humans is virtually non-existent.

Given the sparseness of data on human populations, it is interesting to consider the research of Ridker and colleagues, who explored the benefits of suppressing IL-1beta production with canakinumab in a large, randomized clinical trial. They found reduced cardiovascular events in the treatment arms independent of and separate from lipid reductions by medications such as statins; however, they also found meaningful levels of residual inflammation even after treatment [32,33]. 

They suggest, plausibly in our opinion, that it might be necessary to inhibit inflammation at an earlier stage than IL-1beta, which would be at the level of inflammasomes; for PM, that would mean the NLRP3 inflammasome. Since their findings point to a critical role of inflammasomes in illness secondary to chronic inflammation, we suggest that there is a need for further epidemiological research on PM, including UFPs, and NLRP3 activation in human populations.

A deeper understanding of the pathway by which PM generates a low burn of inflammation would be useful in multiple ways: First, it would add to the science behind the mechanisms by which PM cause adverse health outcomes, further cementing the causal nature of the association for PM_2.5_ and establishing a causal case for UFPs. Second, as with Ridker’s work on canakinumab as a therapeutic agent, it might be that inhibitors of NLRP3 activation could be protective against adverse effects of breathing in ambient PM. 

One possibility is that some traditional Chinese medicines could be protective. There is considerable evidence that these medicines act through NLRP3 [34]. More than a dozen heat-clearing and detoxifying herbs, such as Scutellaria baicalensis, Coptis chinensis, Flos lonicerae, Forsythia suspensa, Radix isatidis, and Houttuynia cordata, have been reported as having anti-inflammatory and antimicrobial functions [35,36]. Besides being ingested as traditional medicine, anti-inflammatory herbs are also commonly made in easily accessible tea bags (e.g., Flos lonicerae, Radix isatidis, gingko, and ginseng) or found in the kitchen (e.g., ginger, Portulaca oleracea, cinnamon, red peppers, orange peel, and Houttuynia cordata), being used very frequently but not necessarily being perceived as herbal medicine [37,38]. 

In addition, some studies show that acupuncture may also have an anti-inflammatory effect [39,40,41,42]. The conscious use of traditional medicine for healing and wellness, or unconscious cultural practices in daily life, may play a protective role. This possibility is supported by data that we collected from Asian immigrants, which suggest reduced immune responses to UFP exposure [5,43].

Ultimately, greater knowledge about how PM activates the NLRP3 inflammasome could help us understand which fractions of PM are the most toxic, which would, in turn, suggest priorities for reducing sources of exposure.

## Figures and Tables

**Figure 1 toxics-11-00202-f001:**
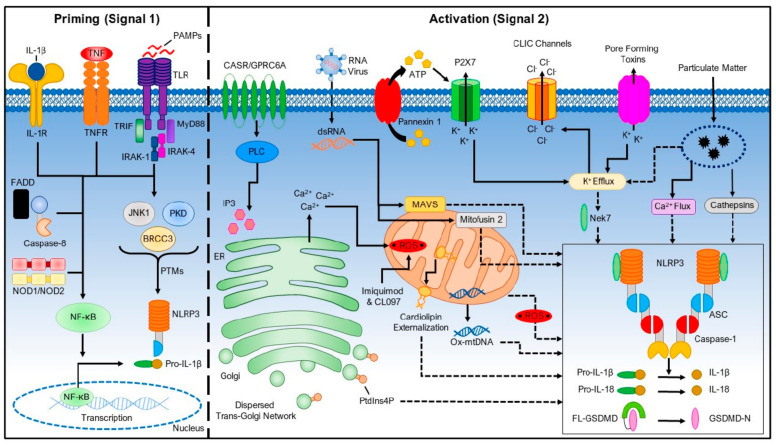
Pathways for NLRP3 inflammasome activation, illustrating priming (**left**) and activation (**right**). The interaction of particulate matter with these pathways is included in the upper-right corner (reproduced from Kelley et al., 2019 [10]).

## Data Availability

Not applicable.

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
