# Peer review of "On the Need for Human Studies of PM Exposure Activation of the NLRP3 Inflammasome"

_toxics, 2023, doi:10.3390/toxics11030202_

Round 1
Reviewer 1 Report
In this comment, Brugge et al. discuss the need for human studies of particulate matter (PM) exposure activation of the NLRP3 inflammasome. The manuscript is interesting and well-written, but I have some requests for the authors:
- Please, add references supporting this sentence: "In both research on UFP exposure and in the broader literature on particulate matter (PM) generally, especially PM2.5 (PM < 2.5 um), there are many reports of association with biomarkers of inflammation such as C-reactive protein (CRP) and Interleukin-6 (IL-6)."
- At the end of the first paragraph, in addition to cardiovascular diseases, it would be interesting to add other diseases potentially triggered by air pollution-related inflammation.
- Please add references supporting information in the second paragraph.
- I strongly suggest adding a figure supporting the main comment ideas and the hypothesis described in the fourth paragraph.
- Please add some references supporting the fifth paragraph, especially the first sentence.
- No need to declare "Paul Ridker and colleagues". I suggest using "Ridker and colleagues" or a similar expression. Also, please add specific reference supporting this sentence: “in a large randomized clinical trial…” (only two references, 27-28, are included at the end of the paragraph).
- Remove acupuncture from the last paragraph. This is still speculative.
Author Response
Reviewer 1:
In this comment, Brugge et al. discuss the need for human studies of particulate matter (PM) exposure activation of the NLRP3 inflammasome. The manuscript is interesting and well-written, but I have some requests for the authors:
Thank you for seeing the value of our commentary.
- Please, add references supporting this sentence: "In both research on UFP exposure and in the broader literature on particulate matter (PM) generally, especially PM2.5 (PM < 2.5 um), there are many reports of association with biomarkers of inflammation such as C-reactive protein (CRP) and Interleukin-6 (IL-6)."
We have added three references here.
- At the end of the first paragraph, in addition to cardiovascular diseases, it would be interesting to add other diseases potentially triggered by air pollution-related inflammation.
Good idea, we now mention respiratory and neurological diseases and added a citation.
- Please add references supporting information in the second paragraph.
We have added a citation.
- I strongly suggest adding a figure supporting the main comment ideas and the hypothesis described in the fourth paragraph.
We have added a figure and cited the source.
- Please add some references supporting the fifth paragraph, especially the first sentence.
We cite these papers in subsequent paragraphs. It might be burdensome to the reader to have them all cited again at this point in the text.
- No need to declare "Paul Ridker and colleagues". I suggest using "Ridker and colleagues" or a similar expression. Also, please add specific reference supporting this sentence: “in a large randomized clinical trial…” (only two references, 27-28, are included at the end of the paragraph).
The two references at the end of the paragraph are the studies by Ridker et al. We use the convention that we cite studies once for the continuous text prior to the citation. If the editors prefer, we could cite at both sentences, but we do not think that is necessary or standard formatting.
- Remove acupuncture from the last paragraph. This is still speculative.
We have added citations that support our understanding that the evidence for acupuncture is strong enough to justify further research.

Reviewer 2 Report
This commentary briefly introduces the toxicity mechanism of NLRP3 inflammasome and the recent progress in the effects of PM2.5 and PM10 on NLRP3 activation in cells and animals. Authors predict the potential toxicity role of NLRP3 inflammasome in human health and suggest future human studies. Some critical points need to be addressed before further consideration.
1. More details should be supplemented to clarify the recent progress of PM on NLRP3 activation.
2. Once NLRP2 inflammasome is activated, what kinds of diseases probably will be caused in human?
3. What is current epidemiological outcome about the relationship between PM and human health?
4. More details should be introduced for the effect of traditional Chinese medicines on inflammation suppression.
Author Response
Reviewer 2:
This commentary briefly introduces the toxicity mechanism of NLRP3 inflammasome and the recent progress in the effects of PM2.5 and PM10 on NLRP3 activation in cells and animals. Authors predict the potential toxicity role of NLRP3 inflammasome in human health and suggest future human studies. Some critical points need to be addressed before further consideration.
We thank the reviewer for their constructive feedback that we think helps us improve our manuscript.
- More details should be supplemented to clarify the recent progress of PM on NLRP3 activation.
We have added a figure with details of how PM interacts with NLRP3 at the cellular level. We have also added:
“It is notable that ambient PM and inflammation activated by NLRP3 are associated with many of the same disease outcomes.”
If there are other aspects of PM and NLRP3 that we are missing that the reviewer wishes to point us to, please advise. We have kept this brief, as is fitting for a commentary, but acknowledge that a full systematic review would be worthwhile, however we cannot undertake that level of work at this time.
- Once NLRP2 inflammasome is activated, what kinds of diseases probably will be caused in human?
We have added to the end of paragraph 4:
“The resulting inflammation could then be associated with a wide range of diseases, including cardiovascular, respiratory and neurological.”
- What is current epidemiological outcome about the relationship between PM and human health?
We have modified the last sentence of the first paragraph, see response to reviewer 1:
“This has led to low level, chronic systemic inflammation becoming a leading hypothesis for the underlying mechanism by which PM influences development of adverse cardiovascular outcomes4 as well as respiratory and neurological diseases.”
- More details should be introduced for the effect of traditional Chinese medicines on inflammation suppression.
We have substantially edited this section and revised the citations accordingly.

Round 2
Reviewer 1 Report
The authors answered most of my requests satisfactorily. I have just one last request: when the authors cite scientific names (e.g. plant, herb names), they must follow the taxonomic rules: Genus (first letter in uppercase) species (all lowercase), both in italics. Popular names may be written in lowercase and without italics. Please review this.
Author Response
Reviewer 1: The authors answered most of my requests satisfactorily. I have just one last request: when the authors cite scientific names (e.g. plant, herb names), they must follow the taxonomic rules: Genus (first letter in uppercase) species (all lowercase), both in italics. Popular names may be written in lowercase and without italics. Please review this.
We have made this correction
Reviewer 2 Report
Authors have addressed all questions.
Author Response
There were no further corrections asked for...